# Erb-b2 Receptor Tyrosine Kinase 2 (ERBB2) Promotes ATG12-Dependent Autophagy Contributing to Treatment Resistance of Breast Cancer Cells

**DOI:** 10.3390/cancers13051038

**Published:** 2021-03-02

**Authors:** Yongqiang Chen, Ruobing Wang, Shujun Huang, Elizabeth S. Henson, Jayce Bi, Spencer B. Gibson

**Affiliations:** 1Research Institute in Oncology and Hematology, Cancer Care Manitoba, University of Manitoba, Winnipeg, MB R3E 0V9, Canada; ychen2@cancercare.mb.ca (Y.C.); wangrb18@mails.jlu.edu.cn (R.W.); huangs12@myumanitoba.ca (S.H.); Elizabeth.Henson@umanitoba.ca (E.S.H.); bij@myumanitoba.ca (J.B.); 2Rady Faculty of Health Sciences, College of Pharmacy, University of Manitoba, Winnipeg, MB R3E 0T5, Canada; 3Department of Biochemistry and Medical Genetics, University of Manitoba, Winnipeg, MB R3E 3N4, Canada

**Keywords:** breast cancer, ERBB2/HER2/Neu, lapatinib, Taxol/paclitaxel, ATG12

## Abstract

**Simple Summary:**

Expression of the tyrosine kinase receptor ERBB2 in cancer cells leads to drug resistance. Autophagy, a “self-eating” process inside the cell, is a mechanism for drug resistance in cancer cells. It has been shown that ERBB2 activation leads to increased autophagy in breast cancer cells, but the underlying mechanisms remains unclear. In this study, we demonstrated that ERBB2 promotes autophagy by increasing the protein levels of the autophagy gene *ATG12* (autophagy-related 12), contributing to the resistance of breast cancer cells to chemotherapy drugs or ERBB2-targeted antibody treatments. We further showed that *ATG12* expression in breast tumors containing ERBB2 correlated with a worse patient survival outcome. Finally, lapatinib is an inhibitor for both EGFR and ERBB2 tyrosine kinases in the EGFR protein family and promotes autophagy in cells containing only EGFR but inhibits autophagy in cells containing only ERBB2. Taken together, this suggests that ERBB2 promotes autophagy through upregulation of ATG12.

**Abstract:**

The epidermal growth factor receptor (EGFR) family member erb-b2 receptor tyrosine kinase 2 (*ERBB2*) is overexpressed in many types of cancers leading to (radio- and chemotherapy) treatment resistance, whereas the underlying mechanisms are still unclear. Autophagy is known to contribute to cancer treatment resistance. In this study, we demonstrate that ERBB2 increases the expression of different autophagy genes including *ATG12* (autophagy-related 12) and promotes ATG12-dependent autophagy. We clarify that lapatinib, a dual inhibitor for EGFR and ERBB2, promoted autophagy in cells expressing only EGFR but inhibited autophagy in cells expressing only ERBB2. Furthermore, breast cancer database analysis of 35 genes in the canonical autophagy pathway shows that the upregulation of *ATG12* and *MAP1LC3B* is associated with a low relapse-free survival probability of patients with ERBB2-positive breast tumors following treatments. Downregulation of ERBB2 or ATG12 increased cell death induced by chemotherapy drugs in ERBB2-positive breast cancer cells, whereas upregulation of ERBB2 or ATG12 decreased the cell death in ERBB2-negative breast cancer cells. Finally, ERBB2 antibody treatment led to reduced expression of ATG12 and autophagy inhibition increasing drug or starvation-induced cell death in ERBB2-positive breast cancer cells. Taken together, this study provides a novel approach for the treatment of ERBB2-positive breast cancer by targeting ATG12-dependent autophagy.

## 1. Introduction

The epidermal growth factor receptor (EGFR) members including EGFR, erb-b2 receptor tyrosine kinase 2 (ERBB2/HER2), ERBB3 and ERBB4, are overexpressed or activation-mutated in many types of cancers, especially in breast, ovarian and non-small cell lung cancers, which is associated with (radio- and chemotherapy) treatment resistance, metastasis, and poor prognosis [1,2]. As the most potent oncogene of the EGFR family members, *ERBB2* is over-expressed in the tumors of many types of cancers such as breast cancer, gastric cancer, colon cancer, bladder cancer, and biliary cancer [3]. In breast cancer, the most common cancer in women worldwide, ERBB2 is overexpressed in about 11–30% of all cases [4,5,6]. Resistance to ERBB2-targeted therapy including the use of trastuzumab, an anti-ERBB2 monoclonal antibody, and lapatinib, a small molecule kinase inhibitor of ERBB2 and EGFR, is associated with alternations of signal transduction pathways, apoptosis, and cell cycle control in breast cancer [7]. The underlying mechanisms are still largely unclear.

Autophagy (macroautophagy) is known to play important roles in therapy resistance in cancers [8,9]. It is an intracellular degradation process characterized by the formation of the double-membraned structure autophagosome [10,11]. During autophagy, cytoplasmic cargos such as dysfunctional organelles and aggregate proteins are enclosed in autophagosomes that will fuse with lysosomes to form autolysosomes where degradation of cargos occurs to generate small molecules being used for biosynthesis and energy sources to support cell survival. However, when autophagy is over-enhanced so that the essential components for cell survival are degraded, it can induce cell death which is called autophagic cell death. Our previous study demonstrated that, when cancer cells were treated under hypoxia, a low level of autophagy promoted cell survival at an early time of hypoxia, whereas an enhanced level of autophagy induced autosis (a type of autophagic cell death) at a later time of hypoxia [12]. Thus, autophagy can be a double-edged sword by either promoting cell survival or cell death depending upon the context [11]. The pro-cell survival mechanism of autophagy contributes to cancer therapy resistance. For example, inhibition of autophagy increased cancer cell death induced by docetaxel [13], and Taxol (paclitaxel) [14,15]. Studies from our and other groups have demonstrated that the activation of EGFR tyrosine kinase can inhibit autophagy by interacting with Beclin 1 [12,16]. ERBB2 has also been reported to inhibit autophagy by interacting with Beclin 1 [17,18]. However, contradictory roles of the ERBB2 monoclonal antibody Trastuzumab (Herceptin) in autophagy have been reported to promote [19] or inhibit autophagy [20]. Therefore, the roles of ERBB2 in regulating autophagy and autophagy in ERBB2-induced treatment resistance need to be further defined.

In this study, we demonstrate that ERBB2 promotes autophagy involving the upregulation of autophagy proteins. Furthermore, ERBB2-induced treatment resistance in breast cancer is contributed to at least by specific upregulation of ATG12 (autophagy-related 12) which promotes autophagy. 

## 2. Materials and Methods

### 2.1. Reagents, Antibodies and Plasmids 

Ammonium chloride (NH_4_Cl) (A9434), chloroquine diphosphate (CQ) (C6628), Sodium orthovanadate (S6508), Taxol (*Paclitaxel)* (T7402), docetaxel (01885), trypan blue solution (T8154), and phosphatase inhibitor cocktails 2 and 3 (P5726, P0044) were purchased from Sigma-Aldrich (Oakville, ON, Canada), protease inhibitor cocktail (11 836 153 001) from Roche Diagnostics (Mannheim, Germany), and Pierce™ Protein G Magnetic Beads (88847) from Thermo Fisher Scientific (Winnipeg, MB, Canada). Lapatinib (L-4899) was purchased from LC Labs (Woburn, MA, USA). 

EBSS Medium for AA (amino acids and serum) starvation (SH30029.02) was purchased from HyClone Laboratories Inc. (Logan, UT, USA).

The Control siRNA-A (si*Con*) (Sc-37007), the siRNAs against *ERBB2* (si*ERBB2*) (sc-29405) and *ATG12* (si*ATG12*) (sc-72578), ERBB2 double nickase plasmid (sc-400138-NIC) (NIC-*ERBB2*) and the control double nickase plasmid (sc-437281) (NIC-*Con*) were purchased from Santa Cruz Biotechnology (Dallas, TX, USA). 

Primary antibodies: anti-EGFR (2232), anti-ERBB2 (2165), anti-ATG12 (4180), anti-ULK1 (8054), anti-FIP200 (12436), anti-BECN1 (3495), anti-ATG5 (2630), anti-ATG7 (2631), anti-LC3B (2775S), anti-SQSTM1/p62 (D1Q5S) (39749), anti-GFP (2956), and rabbit IgG control (2729) antibodies were purchased from Cell Signaling Technology (Whitby, ON, Canada), anti-ATG12 (ab109491) (for immunohistochemistry (IHC)) from Abcam Inc. (Toronto, ON, Canada), and anti-actin beta (ACTB) (A3853) from Sigma-Aldrich (Oakville, ON, Canada). Anti-human ERBB2 antibody (research-grade trastuzumab biosimilar) (MAB9589) was purchased from R&D Systems, Inc. (Minneapolis, MN, USA). Anti-ATG12 (4180) and anti-ATG5 (2630) antibodies mainly identify the ATG12–ATG5 conjugate complex.

Secondary antibodies: goat anti-rabbit IgG (H^+^L)–HRP (horseradish peroxidase) conjugate (170–6515) and goat anti-mouse IgG (H^+^L)–HRP conjugate (170–6516) were obtained from Bio-Rad Laboratories (Montréal, QC, Canada).

EGFR plasmid (#11011) [21], ERBB2 plasmid (#16257) [22], and pmRFP-LC3B (plasmid encoding monomeric red fluorescent protein-tagged LC3B) plasmid (#21075) [23] were purchased from Addgene (Watertown, MA, USA). The empty plasmid vector (EX-NEG-M45) ((3×)HA/Vector) was purchased from GeneCopoeia (Rockville, MD, USA) and was used as a control for transfection with EGFR or ERBB2. GFPSpark-ATG12 plasmid (HG11111-ANG) and its empty plasmid vector (GFPSpark tag empty vector) (CV027) were purchased from Sino Biological (Burlington, ON, Canada). 

### 2.2. Cell Culture

The breast cancer cell lines MDA-MB-231, SKBR3 and MCF7, and the mouse embryonic fibroblast cell line NIH3T3 were grown in Gibco DMEM (Dulbecco’s Modified Eagle Medium), high glucose medium (11965092, Thermo Fisher Scientific (Winnipeg, MB, Canada)) supplemented with 100 units of penicillin per mL plus 100 µg of streptomycin per mL (15140–122, Thermo Fisher Scientific) and 5% fetal bovine serum, in a humidified 5% CO_2_, 37 °C incubator.

### 2.3. Western Blot Analysis and IMMUNOPRECIPITATION (IP)

Western blotting was described previously [24]. Immunoprecipitation (IP) is also similar to that in our previous publication [12]. Briefly, total cell lysate (TCL) was generated by using NP40 (Sigma, I8896) protein lysis buffer [25], with the addition of protease inhibitor cocktail, phosphatase inhibitor cocktails 2 and 3 and sodium orthovanadate. TCL containing 500 µg protein was incubated with anti-ERBB2 or IgG control antibody at 4 °C overnight. Then, 100 μL of Protein G Magnetic Beads was added to the mixture of protein lysate and a primary antibody, followed by incubation at 4 °C overnight. The IP complex was washed 4 times with NP40 lysis buffer. The pellet was re-suspended in protein loading dye and immunoblotting followed. ACTB was used as a loading control. The Image J program (Image J 1.44p, Wayne Rasband, National Institute of Health, Bethesda, MD, USA) was used to quantify the density of western blot protein bands.

### 2.4. Measurement of Autophagy

During autophagy, the cytosolic form of microtubule*-*associated protein 1A/1B-light chain 3 (LC3), LC3-I, is converted to its lipidated form, LC3-II, in autophagosome membranes. Then, LC3-II in the inner membrane of autophagosome will be degraded in the autolysosome. To detect functional autophagy, autophagic flux was measured by Western blotting or fluorescent microscopy. The protein level of LC3-II, in the absence and presence of a lysosomal inhibitor chloroquine (CQ, 20 µM) or ammonium chloride (NH_4_Cl, 30 mM), was measured by Western blotting. An increase in the level of LC3-II in the presence of CQ or NH_4_Cl compared to that in its absence indicates a positive autophagic flux and therefore functional autophagy. Functional autophagy can also be confirmed by Western blotting the protein levels of the autophagy substrate SQSTM1/p62 in the absence and presence of a lysosomal inhibitor (CQ or NH_4_Cl). An increase in the protein level of p62 in the presence of CQ or NH_4_Cl compared to that in its absence correlates with functional autophagy. The third method to measure autophagy is to quantify the number of autophagosomes/autolysosomes by fluorescent microscopy. Cells were transfected with pmRFP-LC3B plasmid, treated and then attached onto a microscope slide by Cytospin centrifugation. Then, the expression of LC3 (red color) was observed with a fluorescent microscope. The diffused red fluorescent color represents LC3-I and the mRFP-LC3 red puncta represent LC3-II-located autophagosomes/autolysosomes. Autophagy was quantified by counting the number of mRFP-LC3 puncta per cell (at least 20 cells were counted for each treatment). CQ was used to measure autophagic flux.

### 2.5. Gene Knockdown by RNA Interference (RNAi)

Knockdown of *ERBB2* or *ATG12* was performed by using siRNAs from Santa Cruz Biotechnology. Control siRNA (si*Con*) and ERBB2 siRNA (si*ERBB2*) or ATG12 siRNA (si*ATG12*) were transfected into cells using Lipofectamine 2000 (11668-019, Invitrogen/Thermo Fisher Scientific). ERBB2 or ATG12 knockdown was confirmed by Western blotting. ERBB2 siRNA and ATG12 siRNA are a pool of three to five target-specific 19–25 nt siRNAs against human *ERBB2* gene and *ATG12* gene, respectively.

### 2.6. Generation of Gene Knock-Out Cell Lines Using CRISPR/Cas9 Gene Editing

The double nickase Clustered Regularly Interspaced Short Palindromic Repeats/CRISPR associated protein 9 (CRISPR/Cas9) plasmids designed by Santa Cruz Biotechnology were used to knock out ERBB2 in cells. ERBB2 double nickase plasmid (NIC-*ERBB2*) and the control double nickase plasmid (NIC-*Con*) were transfected into SKBR3 cells using Lipofectamine 2000. Then, stable cell lines were developed with puromycin selection followed by single colony pickup. ERBB2 knockout was verified by Western blotting and its effect on autophagy was confirmed by comparison experiments with ERBB2 siRNA knockdown. The pair of cell lines (*NIC-Con* cell line and *NIC-ERBB2* cell line) developed from the same parental SKBR3 cell line was developed and maintained parallelly so that the results generated from these two cell lines can be comparable.

### 2.7. Generation of Stable Cell Lines with Gene Overexpression

Empty vectors EX-NEG-M45 and GFPSpark, and EGFR, ERBB2 and GFPSpark-ATG12 plasmids were transfected into MDA-MB-231, MCF7 or NIH3T3 cells using Lipofectamine 2000 and stable cell lines were generated by treating cells with G418 (Geneticin) (for Ex-NEG-M45 and ERBB2), puromycin (for EGFR) or hygromycin (for GFPSpark and GFPSpark-ATG12). Then, a pool of selected cells was used for further experiments. Gene overexpression was confirmed by Western blotting. Each pair of cell lines (vector alone cell line and gene overexpression cell line) developed from the same parental cell line was developed and maintained parallelly so that the results generated from these two cell lines can be comparable. It should be noted that the results generated from a parental cell line and its derived cell line with stable gene expression may not be comparable due to potentially unsynchronized changes of genetic components or signaling pathways during the selection and maintaining process if these two cell lines were not developed and maintained parallelly. This is especially true for the comparison of autophagy levels.

### 2.8. Analysis of Cell Death by Trypan Blue Exclusion Assay

Cell death was measured by flow cytometry as previously described [24] or by counting the percentage of dead cells under microscope [26] after trypan blue staining. Trypan blue is excluded from live cells but penetrates dead cells giving a red fluorescence that can be quantified by flow cytometry. Under a microscope, live cells have a clear cytoplasm but dead cells have a blue cytoplasm following trypan blue staining.

### 2.9. Bioinformatics Analysis of Autophagy Gene Expression in Human Breast Tumor Tissues

The Cancer Genome Atlas (TCGA) Illumina HiSeq 2000 RNA sequencing (RNA-seq) data and clinical information of 1218 breast tumor samples were retrieved from the UCSC (University of California, Santa Cruz) Xena browser (http://xena.ucsc.edu/, accessed on 10 March 2020). The mRNA expression was quantified by RNA-seq by Expectation–Maximization (RSEM) followed by log2 transformation to estimate the gene-level transcription. Clinical immunohistochemistry (IHC) values for HER2 status are available in 776 out of the 1218 breast tumors, where 652 tumors are ERBB2-positive, 114 tumors are ERBB2-negative, and 10 tumors are ERBB2-equivocal. We excluded the 10 ERBB2-equivocal tumors and only analyzed the mRNA expression data of autophagy genes including *ULK1, RB1CC1/FIP200, BECN1, ATG12, ATG5, ATG7,* and *MAP1LC3B* in the remaining 766 breast tumors with ERBB2-positive or -negative. To assess the mRNA expression difference of each autophagy gene between ERBB2-positive and ERBB2-negative groups of tumors, we performed a two-tailed Wilcoxon test to compare the means of gene expression of the two groups and demonstrated gene expression in the two groups using a boxplot.

### 2.10. Bioinformatics Data Extraction of Survival Probability of Breast Cancer Patients

Relapse-free survival (RFS) refers to the time length starting from the primary treatment of cancer in a patient until the time when the patient survives with no signs or symptoms of that cancer. To investigate the upregulation-triggered effects of different autophagy genes on human breast cancer treatment, we extracted the RFS-based survival probability data of patients who have breast cancers with high- and low-levels (mRNA levels) of an autophagy gene associated with ERBB2-positive and -negative statuses in the online Kaplan–Meier Plotter datasets (http://kmplot.com/analysis/index.php?p=service&cancer=breast (accessed on 19 November 2019)) [27]. In the datasets, 252 patients have ERBB2-positive (+) breast tumors and 800 patients have ERBB2-negative (−) breast tumors. The difference between the RFS-based survival probabilities associated with the high-level expression of an autophagy gene and with the low-level expression of this gene is considered statistically significant if a value of *p* < 0.05 is observed. If a low RFS-based survival probability is significantly correlated with the high-level expression of an autophagy gene, we conclude that expression of such a gene in breast tumors may contribute to treatment resistance of breast cancer and therefore the worse outcome (the low survival probability) in patients.

### 2.11. Real-Time PCR for mRNA Quantification

Total RNA was isolated using the Qiagen RNeasy Plus mini kit (Qiagen, 74134) according to the manufacturer’s instructions. mRNAs were first reverse transcribed into cDNA using the Bio-Rad iScript Select cDNA synthesis kit (1708897). To perform qPCR, a cDNA template was used in combination with iTaq Universal SYBR Green Supermix (Bio-Rad, 172-5121). The cycling and data collection were performed on Bio Rad CFX Real-Time Detection System (Bio-Rad, 185-5096) using the supplied software. The primer pairs used for the human *ATG12* gene (Bio-Rad, qHsaCEP0041396) and the housekeeping gene *GAPDH* (Bio-Rad, qHsaCED0038674) were pre-designed by Bio-Rad. The *GAPDH* gene was used to standardize the results. The qPCR reaction was run as follows: 95 °C for 2 min and then 40 cycles of 95 °C for 5 s and 60 °C for 30 s.

### 2.12. Statistical Analysis

All data were generated by at least 3 independent triplicate experiments. Data were represented as means ± standard deviation (SD) (*n* ≥ 3). The Student *t-*test with two-tailed distribution and unequal variances was performed for statistical analysis. A value of *p* < 0.05 is considered to be statistically significant. *, *p* < 0.5; **, *p* < 0.01; ***, *p* < 0.001.

## 3. Results

### 3.1. ERBB2 Promotes Autophagy

Autophagic flux can demonstrate functional autophagy and is measured by Western blotting the autophagy marker protein LC3-II and the autophagy substrate SQSTM1/p62, or by quantifying the number of LC3 puncta containing LC3-II and representing autophagosomes or autolysosomes, without and with the presence of a lysosomal inhibitor, chloroquine (CQ) or ammonium chloride (NH_4_Cl). A positive autophagic flux was exhibited by a higher protein level of LC3-II or p62, or by a larger number of LC3 puncta per cell in the presence of CQ or NH_4_Cl compared to that in its absence. To investigate the role of ERBB2 in autophagy, we knocked out ERBB2 in the breast cancer cell line SKBR3 expressing a high level of ERBB2 by a double nickase-based CRISPR/Cas9 system (Figure 1A). The addition of CQ increased the LC3-II protein level by more than 90% in the control and ERBB2 knockout cells, respectively, increased the p62 protein level by more than 10% in the control and ERBB2 knockout cells, respectively, and increased the number of mRFP-LC3 puncta per cell by more than 60% in the control and ERBB2 knockout cells, respectively (Figure 1A,B and Appendix A), indicating the existence of a positive basal level of autophagic flux. In the presence of CQ, ERBB2 knockout decreased the LC3-II protein level, the p62 protein level, and the number of mRFP-LC3 puncta per cell by 47%, 52%, and 44%, respectively (Figure 1A,B), supporting that ERBB2 knockout inhibited basal autophagy. Under amino acid starvation, CQ addition increased the LC3-II protein level by more than 120% in the control and ERBB2 knockout cells, respectively, and increased the p62 protein level by more than 50% in the control and ERBB2 knockout cells, respectively (Figure 1A); in the presence of CQ, ERBB2 knockout decreased the LC3-II and p62 protein levels by more than 30%, respectively (Figure 1A), supporting that starvation-induced autophagy is functional in both types of cells and is inhibited by ERBB2 knockout. ERBB2 knockout also significantly inhibited amino acid starvation-induced autophagy measured by quantifying mRFP-LC3 puncta per cell (data not shown). Similar results were also observed when ERBB2 was knocked down in SKBR3 cells, where when autophagy was measured by Western blotting LC3-II and p62 in the absence and presence of CQ, siRNA knockdown of ERBB2 decreased both the basal and amino acid starvation-induced levels of autophagy by 10–40%, respectively (Appendix A). On the other hand, when ERBB2 was overexpressed in breast cancer cell lines MDA-MB-231 (ERBB2-negative) and MCF 7 (ERBB2-positive) (Figure 2A) and autophagy was measured by LC3-II Western blotting, basal and amino acid starvation-induced levels of autophagy were increased by at least 40%, respectively (Figure 2B,C). When bafilomycin B1 was used to block lysosomal degradation, similar results were obtained in MCF7 cells (data not shown). Therefore, ERBB2 promotes autophagy.

### 3.2. Lapatinib Increases Autophagy in Cells Expressing EGFR but Inhibits Autophagy in Cells Expressing ERBB2

Previous studies reported that the dual inhibitor of EGFR and ERBB2 tyrosine kinases lapatinib increased autophagy and concluded that ERBB2 inhibited autophagy [17,18,28]. However, the ERBB2-expressing cells used in these studies can also express EGFR. This makes it unclear whether autophagy promotion by lapatinib was due to the inhibition of ERBB2 tyrosine kinase. To clarify this question, we expressed EGFR and ERBB2 in the EGFR family members-negative NIH3T3 cells and observed that lapatinib inhibited activation of EGFR and ERBB2 tyrosine kinases exhibited by phosphorylation at Y1068 and Y877, respectively (Figure 3A, Appendix A). The addition of CQ increased the protein levels of LC3II and p62 in NIH3T3 cells expressing EGFR, vector alone, and ERBB2, respectively (Figure 3B,C), suggesting that functional basal autophagy exists in these three types of cells. EGFR and ERBB2 had opposite effects on basal autophagy where, in the presence of CQ, EGFR expression decreased LC3-II and p62 protein levels by 25% and 34%, respectively, indicating autophagy inhibition; in contrast, ERBB2 expression increased LC3-II and p62 protein levels by 59% and 56%, respectively, indicating autophagy stimulation (Figure 3B). Lapatinib slightly decreased autophagy by 20% in vector alone cells but it increased autophagy by 30% in cells expressing EGFR (Figure 3C), supporting that the inhibition of EGFR tyrosine kinase by lapatinib increased autophagy. In contrast, lapatinib inhibited autophagy by 90% in cells expressing ERBB2 (Figure 3C), indicating that the inhibition of ERBB2 tyrosine kinase by lapatinib decreased autophagy. These results are consistent with the observations in Figure 1 and Figure 2.

### 3.3. ERBB2 Regulates the Expression of Autophagy Genes

To investigate the underlying mechanism for ERBB2 promotion of autophagy, we examined the expression of several essential autophagy genes affected by ERBB2 expression in human breast tumor tissues via bioinformatics analysis of the data in the Oncomine database. ERBB2 expression significantly increases the mRNA levels of Unc-51 like autophagy activating kinase 1(*ULK1)*, autophagy-related 7 (*ATG7*), autophagy-related 5 (*ATG5*) and autophagy-related 12 (*ATG12*), but significantly decreases that of BECN1/Beclin 1(*BECN1*) and has no significant effect on that of RB1 inducible coiled-coil 1 (*RB1CC1*/*FIP200*) (Figure 4A). ERBB2 expression also has no significant effect on the mRNA expression of microtubule associated protein 1 light chain 3 beta (*MAP1LC3B*) (data not shown). When ERBB2 was overexpressed in MDA-MB-231 cells or MCF7 cells (Figure 2A), ULK1, ATG7, ATG5 or ATG12 protein was increased by at least 30% (Figure 4B), whereas when ERBB2 was knocked out in SKBR3 cells (Figure 1A), expression of these autophagy proteins was decreased by at least 50% (Figure 4B), consistent with the change of mRNA levels (Figure 4A). In agreement with the mRNA change (Figure 4A), the BECN1 protein was reduced by at least 20% by ERBB2 overexpression but was increased by 110% by ERBB2 knockout (Figure 4B). Interestingly, FIP200 protein was increased by at least 40% by ERBB2 overexpression but reduced by 80% by ERBB2 knockout (Figure 4B), although its mRNA level is not significantly affected by ERBB2 expression in patient tumor tissues (Figure 4A). Therefore, ERBB2 may upregulate the expression of autophagy proteins to promote autophagy.

### 3.4. ERBB2-Induced Breast Cancer Treatment Resistance Correlates with ATG12 Upregulation and Autophagy Promotion

To investigate whether any of the ERBB2-upregulated autophagy proteins ULK1, FIP200, ATG7, ATG5 and ATG12 (Figure 4B), are required for ERBB2-regulated autophagy and ERBB2-induced treatment resistance, we first performed bioinformatics analysis. In cancer clinical studies, relapse-free survival (RFS) is used to evaluate the efficacy of a treatment. To investigate the upregulation-triggered effects of different autophagy genes on human breast cancer treatment, we extracted the RFS-based survival data of patients who have breast cancers with high- and low-levels of an autophagy gene associated with ERBB2-positive and -negative statuses in the Kaplan–Meier Plotter datasets. Interestingly, we found that, among the 35 autophagy genes analyzed, upregulation of only *ATG12* or *MAP1LC3B* in ERBB2-positive breast tumors leads to lower RFS-based patient survival probability, whereas *ATG12* or *MAP1LC3B* upregulation has no effect on the survival probability of patients with ERBB2-negative breast tumors, in contrast to the cases with upregulation of *BECN1* and other genes (Figure 5 and Appendix A). Since *MAP1LC3B* coded protein LC3B has two forms including LC3B-I and LC3B-II and the levels of LC3B-I and LC3B-II are dynamically changed during a functional autophagy process in ERBB2-negative (MAD-MB-231) and ERBB2-positive (SKBR3 and MCF7) cells (Figure 1, Figure 2 and Figure 3), we chose to further study the roles of ATG12 in ERBB2-regulated autophagy and treatment resistance in breast cancer cells. Immunohistochemistry (IHC) data showed that ATG12 protein is expressed in human breast tumor tissues (Appendix A). In MDA-MB-231 cells, ERBB2 overexpression upregulated the *ATG12* mRNA level 1.7-fold (Appendix A). This is consistent with the upregulation of ATG12 protein in the form of ATG12–ATG5 conjugate by ERBB2 overexpression in MDA-MB-231 and MCF7 cells and the downregulation of ATG12 protein by ERBB2 knockout in SKBR3 cells (Figure 4B). When ATG12 was knocked down in SKBR3 cells (Figure 6A), the basal level of autophagy was inhibited by 40% and amino acid starvation-induced autophagy was inhibited by 50% (Figure 6B). On the other hand, ATG12 overexpression increased basal and amino acid starvation-induced levels of autophagy by 30% and 50%, respectively, in MDA-MB-231 cells (Figure 6A,C). These data support that ERBB2 upregulates ATG12 to promote autophagy.

It is known that cancer treatment resistance can be caused by the pro-cell survival role of autophagy [29,30,31] and by ERBB2 expression [32,33,34]. To examine whether ERBB2 promotion of autophagy leads to treatment resistance of breast cancer cells, we treated the breast cancer cell lines with the chemotherapeutic drug Taxol (paclitaxel). In SKBR3 cells, ERBB2 knockout decreased Taxol-induced autophagy by 80% but increased Taxol-induced cell death by 34% (Figure 7A,B), and Taxol-induced cell death was increased by 36% by ATG12 knockdown (Figure 7B). On the other hand, in MDA-MB-231 cells, ERBB2 overexpression increased Taxol-induced autophagy by 60% but decreased Taxol-induced cell death by 45% (Figure 7C,D), and Taxol-induced cell death was decreased by 43% by ATG12 overexpression (Figure 7D). Taxol-induced cell death in SKBR3 cells was also increased by 59% by the autophagy inhibitor NH_4_Cl or CQ (Appendix A). Cell death induced by another chemotherapeutic drug docetaxel was also increased by 34% by ERBB2 knockout in SKBR3 cells but was decreased by 34% by ERBB2 overexpression in MDA-MB-231 cells (Appendix A). Furthermore, hypoxia-induced cell death was increased by 22% by ERBB2 knockout in SKBR3 cells but was decreased by 27% by ERBB2 overexpression in MDA-MB-231 cells (Appendix A). Therefore, ERBB2 promotes ATG12 expression to increase autophagy leading to resistance of breast cancer cells to cell death induced by a chemotherapeutic drug or hypoxia.

### 3.5. ERBB2 Antibody Induces ATG12 Downregulation and Autophagy Inhibition Contributing to Breast Cancer Treatment Sensitivity

One current therapy for ERBB2-positive breast cancer is to use the anti-ERBB2 monoclonal antibody trastuzumab (Herceptin) [35]. When an anti-human ERBB2 antibody, a research-grade trastuzumab biosimilar, was used to treat SKBR3 cells for 24 h, the ERBB2 protein level was reduced by 60% in cells with complete medium and by 40% in cells with amino acid starvation (Figure 8A). This is consistent with previous studies showing that degradation of ERBB2 protein is a feature for anti-ERBB2 antibody treatment of ERBB2-positive cells [36,37,38,39]. In SKBR3 cells, ERBB2 antibody treatment also led to a decrease in ATG12 protein by 20% in the cells without and with amino acid starvation, respectively (Figure 8A) and a reduction in *ATG12* mRNA level by 17% (Figure 8B). Furthermore, immunoprecipitation data showed that ERBB2 can interact with ATG12 in its ATG12–ATG5 conjugate form (Figure 8C). These data suggest that the downregulation of ATG12 protein induced by ERBB2 antibody treatment could be caused by the reduction in *ATG12* mRNA level and the degradation of ATG12 protein. ERBB2 antibody treatment decreased the basal and amino acid starvation-induced autophagy levels by 60%, respectively (Figure 8D) but increased cell death induced by taxol by 40% in 24 h and cell death without and with amino acid starvation by 180% and 127%, respectively, in 72 h, in SKBR3 cells (Figure 8E). This is consistent with the observation that the ERBB2 antibody Herceptin increased the tyrosine kinase activity of EGFR [38] which leads to autophagy inhibition [12]. Thus, ERBB2 antibody-induced autophagy inhibition may be caused by the degradation of ERBB2, downregulation of ATG12, and increased activation of EGFR tyrosine kinase. These observations support that ERBB2 antibody treatment sensitizes ERBB2-positive breast cancer cells to cell death by inhibiting autophagy. Therefore, an important mechanism for ERBB2-targeting antibody therapy/immunotherapy may be due to autophagy inhibition.

## 4. Discussion

Autophagy plays important role in various human diseases such as cancer, neurodegenerative diseases and microbial infections [40]. This triggers strong interest in investigating the regulation of autophagy for disease management and treatment by targeting autophagy. EGFR signaling plays key role in mammalian cell biology and many human diseases [41,42,43,44,45,46,47].

Among the EGFR family members, EGFR and ERBB2 were demonstrated to regulate autophagy [1,20]. However, contradictory conclusions have been reported for the role of ERBB2 in autophagy regulation [17,18,20]. In this study, we showed that ERBB2 promotes autophagy at least partially by upregulating the expression of ATG12. Furthermore, this promotion of autophagy contributes to the resistance of ERBB2 expression to cell death induced by stresses such as chemotherapy drug treatment in breast cancer cells, which provides insight into proper strategies for the treatment of cancers expressing ERBB2 by targeting autophagy.

Studies have reported that ERBB2 inhibited autophagy and interacted with the autophagy protein Beclin 1 [17,18]. The conclusion of ERBB2 inhibition of autophagy was drawn mainly based on the observation that lapatinib, an inhibitor for both EGFR and ERBB2 tyrosine kinases, increased autophagy in ERBB2-expressing breast cancer cells [17,18,28]. In this study, we clearly demonstrate that ERBB2 promotes autophagy. Furthermore, we distinguished the effects of lapatinib on autophagy regulated by EGFR and ERBB2: lapatinib inhibits EGFR leading to autophagy increase, whereas it inhibits ERBB2 leading to autophagy inhibition. Our findings are consistent with the report that the ERBB2 specific antibody Herceptin (Trastuzumab) inhibits autophagy [20]. Since the ERBB2-expressing cancer cell lines, BT474, SKBR3, MDA-MB-361, MCF7, and OE19, used by the other studies [17,18,28] also express EGFR [48,49,50], the lapatinib-induced autophagy promotion reported by these studies may be due to lapatinib inhibition of EGFR. Lapatinib-induced inhibition of ERBB2-mediated autophagy may contribute to lapatinib-induced cell death. This is supported by the study that the development of lapatinib resistance of ERBB2-positive breast cancer cells is correlated with upregulation of autophagy where the addition of autophagy inhibitors reversed lapatinib resistance by inhibiting cell proliferation and increasing cell death [51].

Autophagy plays dual roles in cancer. The physiological levels of autophagy in normal cells can prevent the initiation of cancer possibly by suppressing genomic instability [52]. In contrast, in established cancer cells, basal levels of autophagy promote cell survival and, when autophagy is over-enhanced, it can induce autophagic cell death [31,53,54]. Promotion of autophagy can be induced by overexpression of an autophagy gene such as *BECN1* [55], *ATG3* [56], *ATG5* [57], *ATG7* [58], and *ATG8* [59]. Our bioinformatic analysis of published datasets shows that ERBB2 expression in human breast cancer tissues leads to upregulation of several essential autophagy genes including *ATG12* (Figure 4). We further confirmed that ERBB2 expression increased the mRNA and protein levels of *ATG12* contributing to autophagy increase. Hence, ERBB2 increasing autophagy is at least partially due to its upregulation of ATG12. Furthermore, we analyzed 35 autophagy genes involved in the classical autophagy pathway and found that *ATG12* and *MAP1LC3B* are the only genes which upregulation is correlated with worse RFS-based survival probability in patients with ERBB2-positive but not ERBB2-negative breast cancer (Figure 5 and Appendix A). This supports that autophagy promotion by upregulation of ATG12 and MAP1LC3B might cause treatment resistance of ERBB2-positive breast cancer, leading to a worse outcome for breast cancer patients.

Increased levels of autophagy in cancer cells can promote cell survival or cell death under stress conditions such as drug treatment, depending upon the context [11,31]. The pro-cell survival role of autophagy contributes to treatment resistance of cancer [29,30,31]. It is well-known that ERBB2 expression can cause chemo- and radio-resistance in breast cancer [32,33,34]. In this study, we demonstrate that autophagy promotion by ERBB2 expression protects breast cancer cells from cell death induced by stresses such as chemotherapy drug treatment. This can be at least the partial mechanism for treatment resistance of ERBB2-positive breast cancer contributing to worse survival rates in ERBB2-positive breast cancer patients. In this study, expression of ERBB2 led to autophagy promotion and resistance to cell death in triple-negative breast cancer cells (lack of ERBB2, estrogen (ER) and progesterone (PR)), supporting that in established cancer cells, the level of autophagy correlates with treatment resistance of cancer. However, the upregulation of autophagy by ERBB2 expression in cancer may provide a therapeutic window for cancer where autophagy can be targeted as it is demonstrated in this study that ERBB2 antibody inhibited autophagy to sensitize ERBB2-positive breast cancers to chemotherapy drugs. It might be true that low levels of basal autophagy exist in triple negative breast cancer making it hard to treat and contributing to its most aggressive feature among different subtypes of breast cancer. This needs to be explored in future studies.

## 5. Conclusions

In this study, we demonstrate that ERBB2 promotes autophagy by upregulating ATG12 as well as other autophagy-related (ATG) proteins including ULK1, FIP200, ATG5, and ATG7, leading to breast cancer treatment resistance and worse outcome to patients with ERBB2-positive breast cancer (Figure 9). One mechanism for the current clinical treatment of ERBB2-positive breast cancer with the ERBB2 antibody Herceptin/Trastuzumab (and possibly others) may be due to autophagy inhibition by the antibody-induced ERBB2 degradation and ATG12 downregulation. Our study provides insight into the right strategies to treat ERBB2-positive cancer via targeting autophagy.

## Figures and Tables

**Figure 1 cancers-13-01038-f001:**
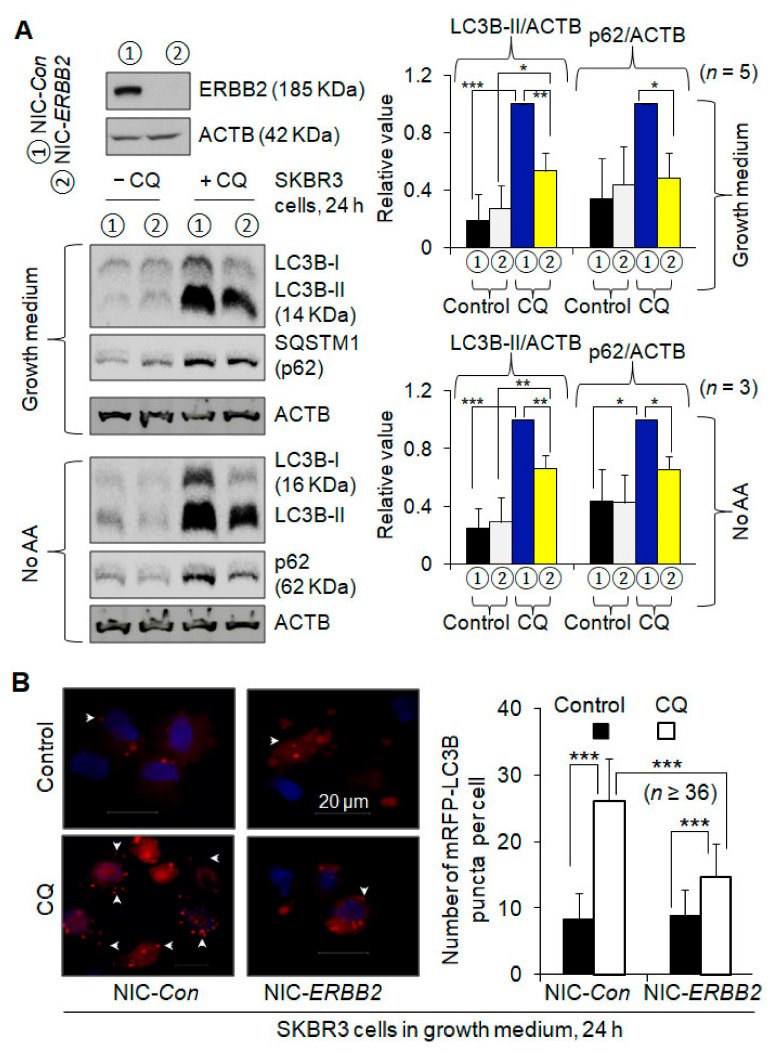
ERBB2 knockout inhibits autophagy in ERBB2-positive SKBR3 breast cancer cells. (**A**) Knockout of ERBB2 was performed by a double-nickase Clustered Regularly Interspaced Short Palindromic Repeats/CRISPR associated protein 9 (CRISPR/Cas9) system and demonstrated by Western blotting. Autophagy was measured by Western blotting LC3-II and p62. Chloroquine (CQ, 20 µM) was used to block lysosomal degradation for autophagic flux measurement (the same hereafter). No AA, amino acid starvation. Anti-actin beta (ACTB) was used as a loading control (the same hereafter). Uncropped Western blot images and quantification of the protein band intensities were shown in Appendix A. (**B**) Autophagy was measured by demonstrating the mRFP-LC3 puncta in cells and counting the numbers of mRFP-LC3 puncta per cell. Data represent or demonstrate the results from at least three independent experiments (the same hereafter). More example images of mRFP-LC3 puncta are shown in the Appendix A. *, *p* < 0.5; **, *p* < 0.01; ***, *p* < 0.001 (the same hereafter). NIC-*Con*, control cells; NIC-*ERBB2*, ERBB2 knockout cells.

**Figure 2 cancers-13-01038-f002:**
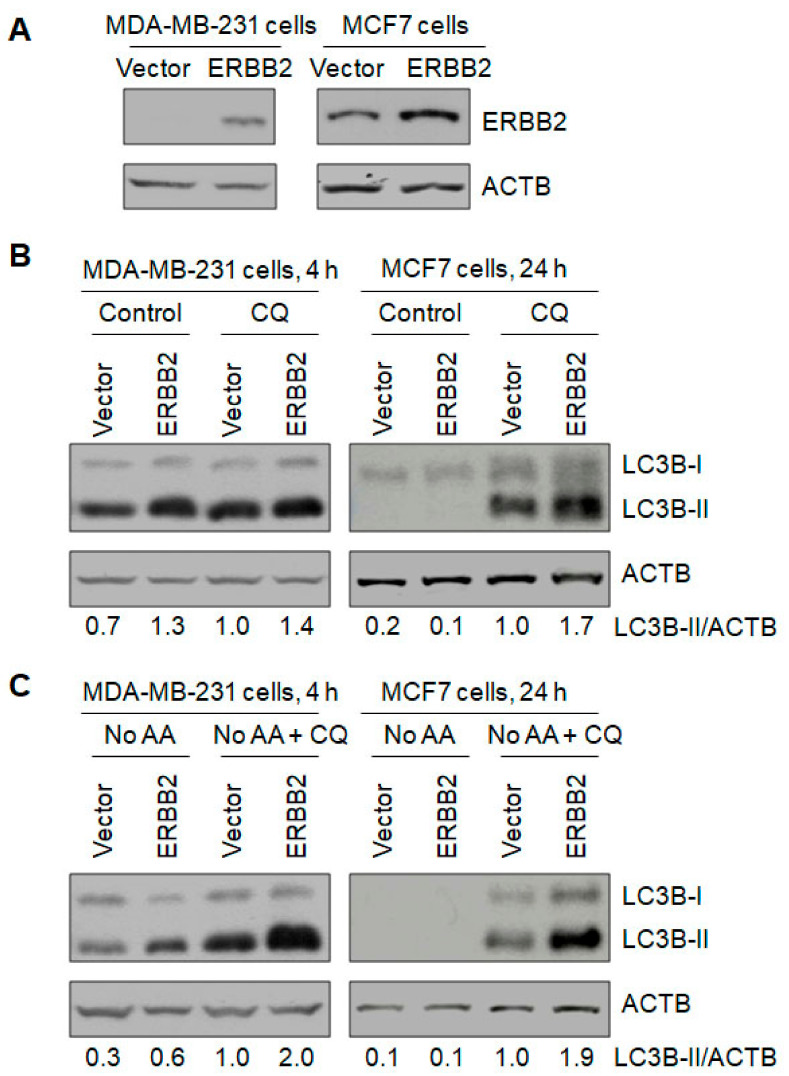
Overexpression of ERBB2 promotes autophagy in ERBB2-negative and -ERBB2-positive breast cancer cells. Western blotting demonstration of ERBB2 overexpression in the ERBB2-negative MDA-MB-231 cells and the ERBB2-positive MCF7 cells (**A**). ERBB2 overexpression increased the basal levels of autophagy (**B**) and amino acid starvation (No AA)-induced levels of autophagy (**C**). Autophagy was measured by Western blotting LC3-II in the absence and presence of CQ. Uncropped Western blot images and quantification of the protein band intensities were shown in Appendix A.

**Figure 3 cancers-13-01038-f003:**
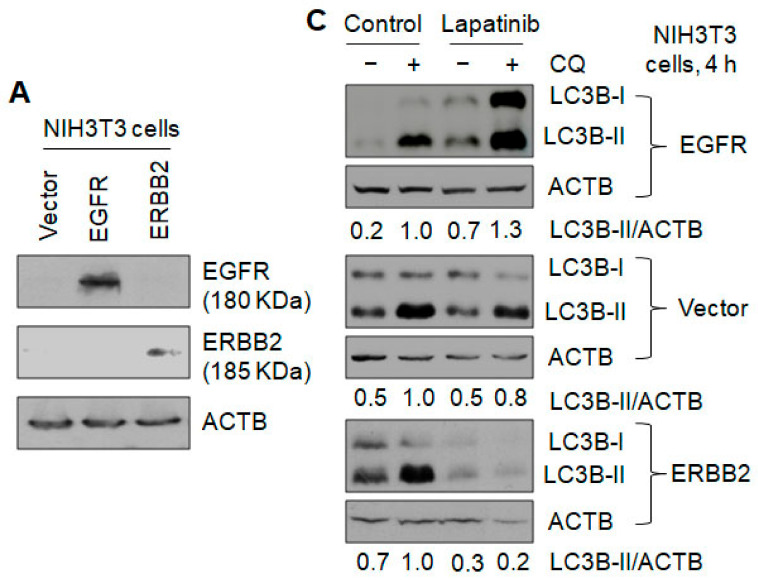
Effects of lapatinib on autophagy in NIH3T3 cells expressing EGFR or ERBB2. (**A**) Western blotting showed the expression of EGFR and ERBB2. (**B**) Effects of EGFR and ERBB2 on basal autophagy. (**C**) Effects of lapatinib (1 µM) on autophagy in cells expressing vector alone, EGFR or ERBB2. Autophagy was measured by Western blotting LC3-II and SQSTM1/p62 in the absence and presence of CQ. Uncropped Western blot images and quantification of the protein band intensities were shown in Appendix A. *, *p* < 0.5; **, *p* < 0.01; ***, *p* < 0.001.

**Figure 4 cancers-13-01038-f004:**
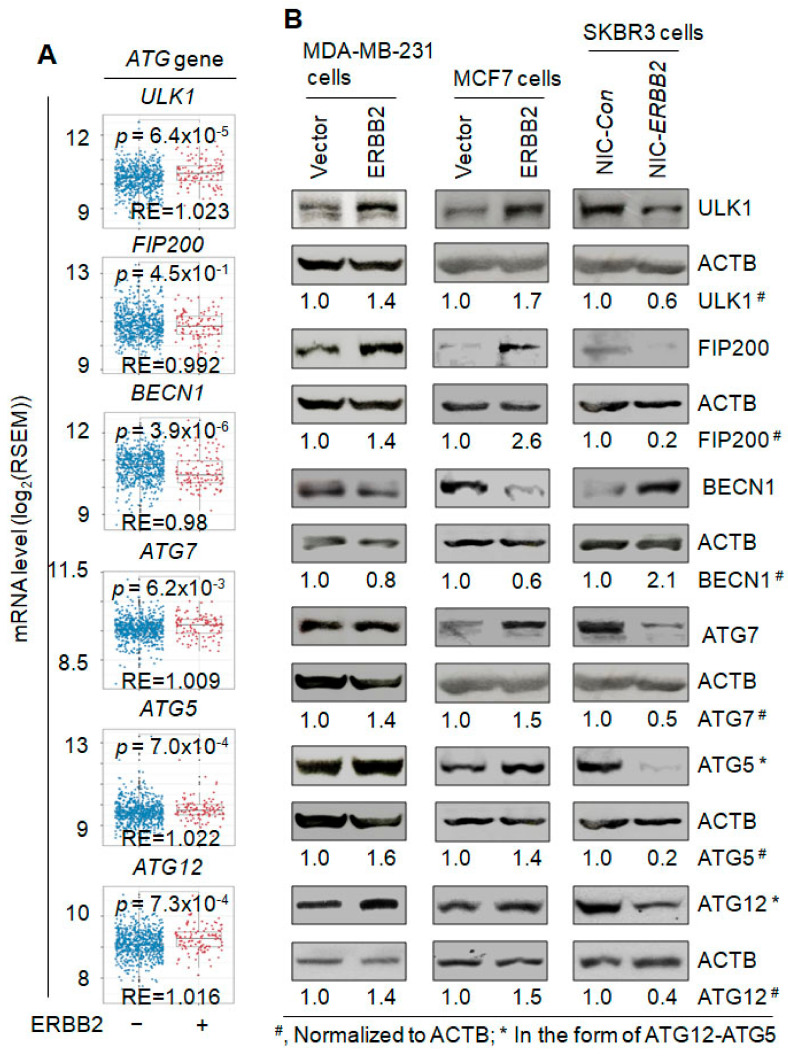
ERBB2 regulates the expression of autophagy genes. (**A**) Bioinformatics analysis of the mRNA levels of different autophagy genes (ATG genes) in ERBB2-positive and -negative human breast tumor tissues. The data were extracted from the Cancer Genome Atlas (TCGA) database. RE, relative mRNA expression of an autophagy gene in ERBB2-positive breast tumors compared to that in ERBB2-negative breast tumors. The mRNA expression was quantified by RNA-seq by Expectation–Maximization (RSEM) followed by log2 transformation to estimate the gene-level transcription. (**B**) Western blotting demonstration of the effects of ERBB2 overexpression and knockout on protein expression of autophagy proteins ULK1, FIP200 (RB1CC1), BECN1, ATG7, ATG5, and ATG12. Overexpression of ERBB2 in MDA-MB-231 and MCF7 cells and ERBB2 knockout in SKBR3 cells are demonstrated in Figure 2A and Figure 1A, respectively. Expression of ATG5 or ATG12 was demonstrated by the level of ATG12–ATG5 complex since it is well known that ATG5 and ATG12 form a conjugate in the form of ATG12–ATG5 during autophagy. Uncropped Western blot images and quantification of the protein band intensities are shown in Appendix A. NIC-*Con*, control cells; NIC-*ERBB2*, ERBB2 knockout cells.

**Figure 5 cancers-13-01038-f005:**
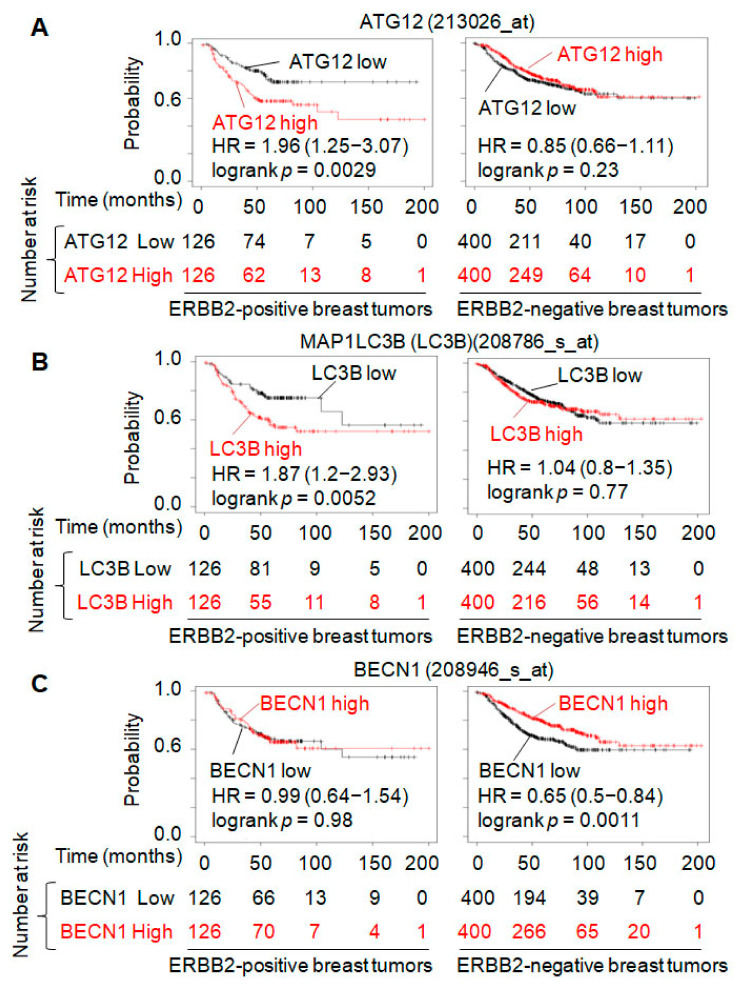
The Kaplan–Meier plots of the relapse-free survival (RFS)-based survival probability of patients with ERBB2-positive and -negative breast tumors. The plots were downloaded from the Kaplan–Meier Plotter website (http://kmplot.com/analysis/index.php?p=service&cancer=breast (accessed on 19 November 2019)). (**A**) Data of patients with ATG12 expression (*ATG12* mRNA) in breast tumors. (**B**) Data of patients with MAP1LC3B (LC3B) expression (*MAP1LC3B* mRNA) in breast tumors. (**C**) Data of patients with BECN1 expression (*BECN1* mRNA) in breast tumors.

**Figure 6 cancers-13-01038-f006:**
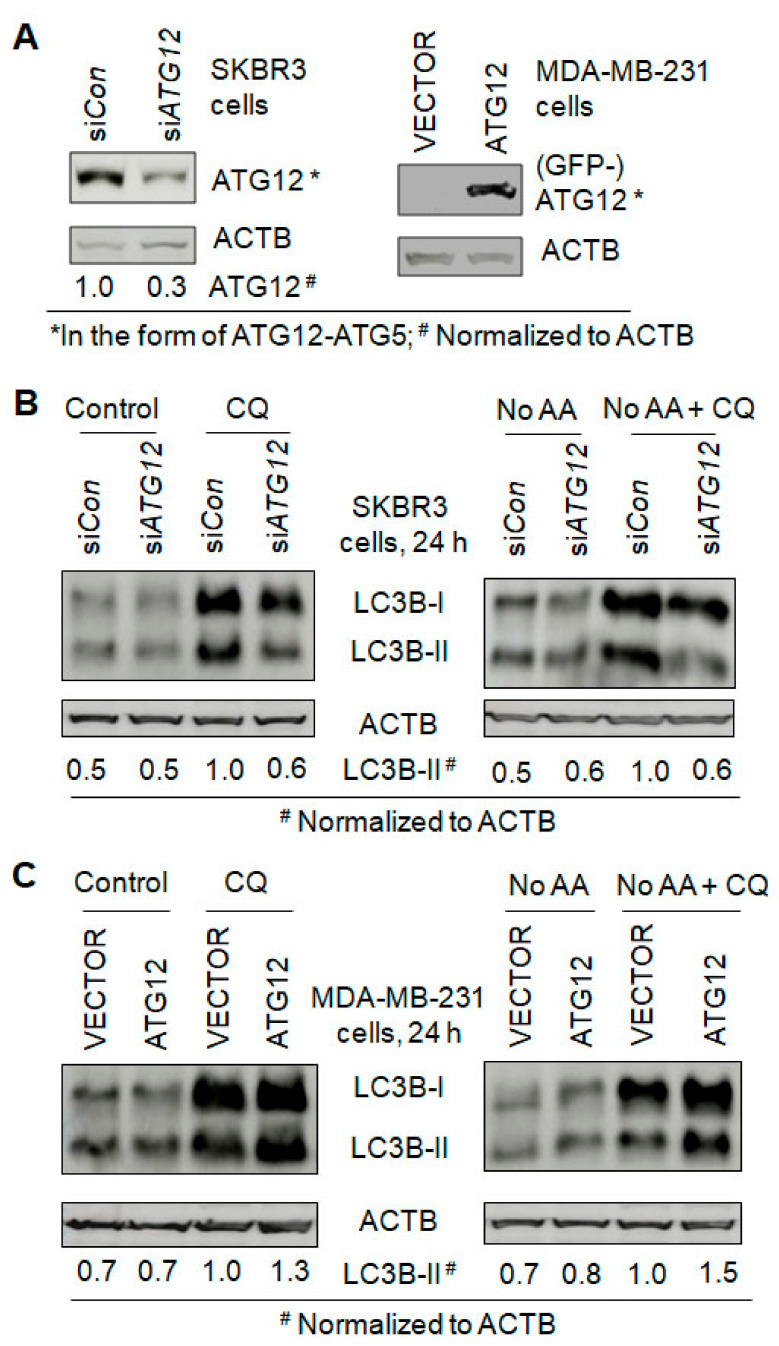
Autophagy is ATG12-dependent in ERBB2-positive and -negative breast cancer cells. Autophagy was measured by Western blotting LC3-II in the absence and presence of CQ. (**A**) Western blotting demonstration of siRNA knockdown of ATG12 in the ERBB2-positive SKBR3 breast cancer cells and ATG12 overexpression in the ERBB2-negative MDA-MB-231 breast cancer cells. (**B**) ATG12 knockdown decreased the basal level and amino acid starvation (No AA)-induced level of autophagy in SKBR3 cells. (**C**) ATG12 overexpression increased the basal level and amino acid starvation (No AA)-induced level of autophagy in MDA-MB-231 cells. Uncropped Western blot images and quantification of the protein band intensities are shown in Appendix A.

**Figure 7 cancers-13-01038-f007:**
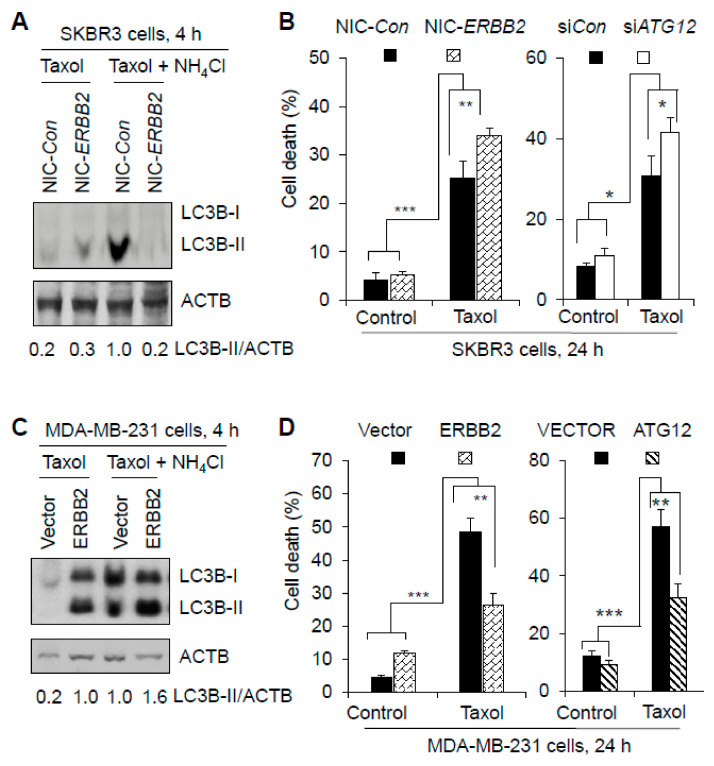
ERBB2 promotion of autophagy inhibits cell death induced by chemotherapy drug Taxol in breast cancer cells. (**A**) ERBB2 knockout decreased autophagy induced by Taxol (paclitaxel, 10 µM) in SKBR3 cells. Autophagy was measured by Western blotting LC3-II in the absence and presence of NH_4_Cl (30 mM). ERBB2 knockout is demonstrated in Figure 1A. (**B**) Cell death induced by Taxol was increased by ERBB2 knockout or ATG12 knockdown in SKBR3 cells. Knockdown of ATG12 and the related effect on autophagy are demonstrated in Figure 6A,B. (**C**) ATG12 overexpression increased Taxol-induced autophagy in MDA-MB-231 cells. Overexpression of ERBB2 is demonstrated in Figure 2A. Autophagy was measured by Western blotting LC3-II in the absence and presence of NH_4_Cl. (**D**) Cell death induced by Taxol was decreased by overexpression of ERBB2 or ATG12 in MDA-MB-231 cells. ATG12 overexpression and the related effect on autophagy are demonstrated in Figure 6A,C. NH_4_Cl, instead of CQ, was chosen to measure Taxol-induced autophagy because CQ can induce higher cell death than NH_4_Cl (Appendix A and unpublished data). Uncropped Western blot images and quantification of the protein band intensities were shown in Appendix A. NIC-*Con*, control cells; NIC-*ERBB2*, ERBB2 knockout cells. *, *p* < 0.5; **, *p* < 0.01; ***, *p* < 0.001.

**Figure 8 cancers-13-01038-f008:**
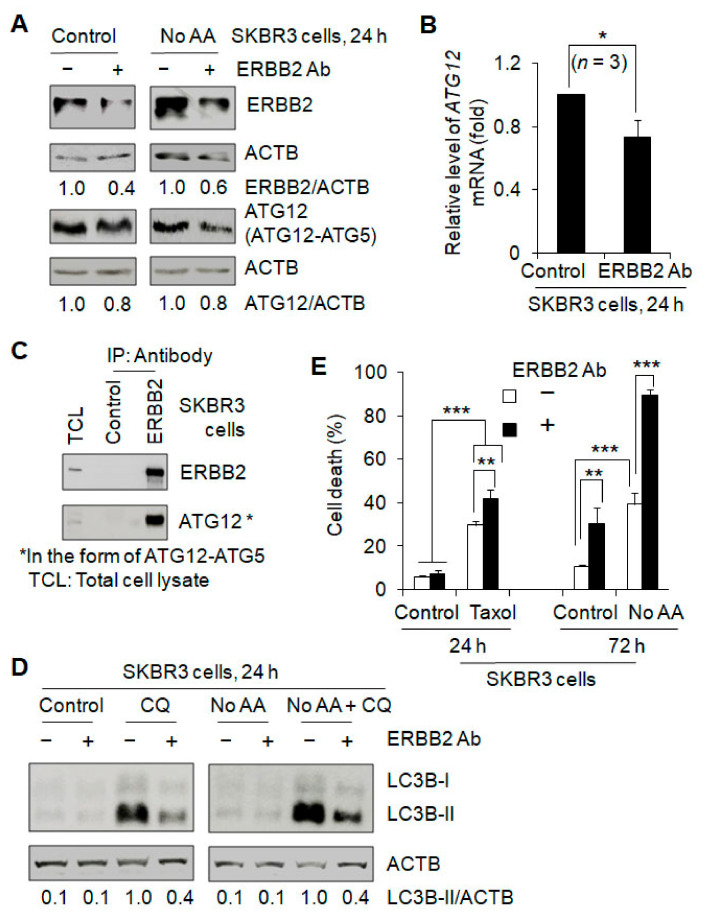
Effects of ERBB2 antibody on ATG12 expression, autophagy, and cell death. An anti-human ERBB2 antibody (Ab) (research-grade trastuzumab biosimilar) was used to treat SKBR3 cells. (**A**) ERBB2 Ab treatment reduced the protein levels of ERBB2 and ATG12. (**B**) Real-time PCR experiments showed that ERBB2 Ab treatment decreased *ATG12* mRNA level. (**C**) Immunoprecipitation (IP) demonstration of the interaction between ERBB2 and ATG12. (**D**) ERBB2 Ab treatment inhibited autophagy at the basal and amino acid starvation (No AA)-induced levels. Autophagy was measured by Western blotting LC3-II in the absence and presence of CQ. (**E**) ERBB2 Ab treatment increased cell death without and with Taxol treatment. Uncropped Western blot images and quantification of the protein band intensities are shown in Appendix A. *, *p* < 0.5; **, *p* < 0.01; ***, *p* < 0.001.

**Figure 9 cancers-13-01038-f009:**
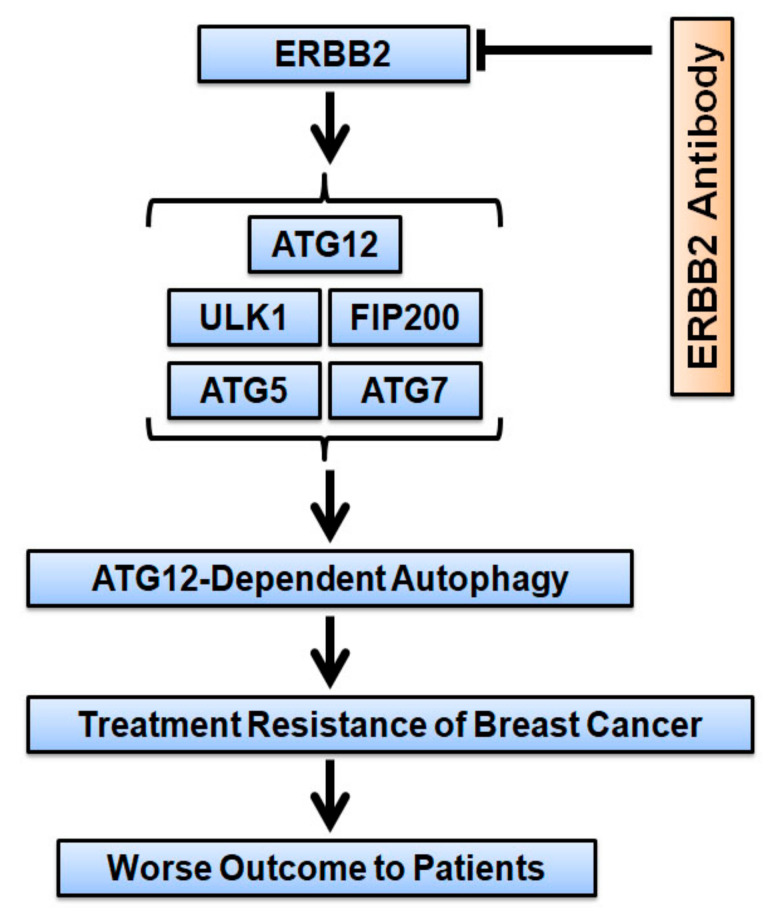
A graphic model for ERBB2 promotion of autophagy and its effect on cancer treatment and outcome to patients. ERBB2 expression upregulates the protein levels autophagy-related (ATG) proteins ULK1, FIP200, ATG5, ATG7, and ATG12 increasing ATG12-dependent autophagy. This leads to treatment resistance of breast cancer and a worse outcome for patients with ERBB2-positive breast cancer. Treatment with ERBB2 antibody (Herceptin or others) causes degradation of ERBB2 and downregulation of ATG12 resulting in autophagy inhibition.

## Data Availability

The data presented in this study are available in this article (and Appendix A).

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
