# Peer review of "Erb-b2 Receptor Tyrosine Kinase 2 (ERBB2) Promotes ATG12-Dependent Autophagy Contributing to Treatment Resistance of Breast Cancer Cells"

_cancers, 2021, doi:10.3390/cancers13051038_

Round 1

Reviewer 1 Report

The authors have fulfilled all requested revisions, therefore this paper is now acceptable for publication in Cancers.

Author Response

Thank you for accepting the manuscript for publication.

Reviewer 2 Report

The respected team has provided all answers to my coments. 

Author Response

Thank for accepting the manuscript for publication.

Reviewer 3 Report

The authors have addressed most of the issues raised in my previous comments. However, a few issues still require authors' attention.

1) The newly added Simple Summary needs thorough English revision. For example:

  • In the statement “Expression of the ERBB2 protein in cancer leads to treatment resistance”, it is not clear to what resistance the authors are alluding to.
  • The statement: “… ATG12 and MAP1LC3B are correlated with high resistance of cancer treatment” is again unclear and inaccurate.
  • The frase: “survival outcome of patients with breast tumors containing ERBB2” is confusing; what do the authors mean by “tumors containing ERBB2”? HER2+ (amplified) or also luminal types expressing ERBB2?
  • Check grammar and typos. For example, in this statement “This suggests that EGFR and ERBB2 plays opposite roles in autophagy” needs to be corrected (“play” instead of “plays”).

2) The “NIC-ERBB2” labeling used (e.g. in Figure 1) to indicate ERBB2 knock-out cells is potentially confusing with cDNA transfer conditions. I recommend spelling out its significance in Figure Legend.

3) It is understood that ATG12 and ATG5 usually form a complex during autophagy, but it is not clear to this reviewer how this complex is recognizable in the presented data? For instance, separate antibody immunoblotting are shown for ATG5 and ATG12 in Figure 4 . Thus, how do the authors know that these proteins are found “in form of an ATG12-ATG5 complex” as indicated in the figure and in the respective legend?  Please explain.

Author Response

Response to Reviewers Comments:

1) The newly added Simple Summary needs thorough English revision. For example: In the statement “Expression of the ERBB2 protein in cancer leads to treatment resistance”, it is not clear to what resistance the authors are alluding to.

Response: "treatment resistance" refers to "conventional therapy resistance (including radiotherapy and chemotherapy)" based on references (1,2) (see Lines 46-47). This has been updated as "(radio- and chemotherapy) treatment resistance" in Lines 13-14, 25-26, and 46, based on the reviewer's suggestion.

2) The statement: “… ATG12 and MAP1LC3B are correlated with high resistance of cancer treatment” is again unclear and inaccurate. The frase: “survival outcome of patients with breast tumors containing ERBB2” is confusing; what do the authors mean by “tumors containing ERBB2”? HER2+ (amplified) or also luminal types expressing ERBB2?

Response: This has been updated as " Analysis of 35 autophagy-related (ATG) genes in a breast cancer database shows that high levels of two ATG genes, ATG12 and MAP1LC3B, in breast tumors containing ERBB2, are correlated with a worse patient survival outcome." Based on the information in the data base (at http://kmplot.com/analysis/index.php?p=service&cancer=breast) and the related reference (Reference 27) (Gyorffy, B.et al. Breast Cancer Res. Treat. 2010, 123, 725-731), "ERBB2/HER2 status" includes "HER2 positive" and "HER2 negative". Therefore, "breast tumors containing ERBB2" should include Group 2 (luminal B) (ER positive, PR negative and HER2 positive) and Group 3 (HER2 positive) (ER negative, PR negative and HER2 positive) breast cancer tumors (please see classification at https://www.mayoclinic.org/diseases-conditions/breast-cancer/in-depth/breast-cancer/art-20045654). Since "Simple Summary" was written so that "layman" readers can understand the paper summary, we hope the updated sentence can satisfy this purpose. Again, we thank the reviewer for pointing out this issue in the previous version of the manuscript.

3) Check grammar and typos. For example, in this statement “This suggests that EGFR and ERBB2 plays opposite roles in autophagy” needs to be corrected (“play” instead of “plays”).

Response: This has been corrected (Line 23). And we have checked grammar and typos in the whole manuscript.

Lines 32-33, " with low relapse-free survival " was changed to "with a low relapse-free survival".

Line 75 " involving upregulation" was changed to " involving the upregulation".

Line 140, " to quantify number of autophagosomes/autolysosomes" was changed to " to quantify the number of autophagosomes/autolysosomes".

Lines 168-169, " Then, a pool of selected cells were used" was changed to " Then, a pool of selected cells was used".

Line 171, " a same parental cell line" was changed to " the same parental cell line".

Line 180, " penetrates into dead cells" was changed to " penetrates dead cells".

Line 186, "(http://xena.ucsc.edu/).The mRNA" was changed to "(http://xena.ucsc.edu/). The mRNA"

Line 193, " To assess mRNA expression" was changed to " To assess the mRNA expression".

Line 198-199, " from a primary treatment of a cancer" was changed to " from the primary treatment of cancer "

Lines 205- 206, " Difference between" was changed to " The difference between ".

Line 207, " considered as statistically" was changed to " considered statistically".

Line 214, " mRNAs was first reverse" was changed to " mRNAs were first reverse".

Line 215, " Rad iScript select cDNA synthesis kit" to " Rad iScript Select cDNA synthesis kit".

Line 217, " on a Bio Rad CFX Real-Time Detection System" to " on Bio Rad CFX Real-Time Detection System".

Line 218, " for human ATG12 gene" to " for the human ATG12 gene".

Line 311, "3.3. ERBB2 regulates expression of autophagy genes" to "3.3. ERBB2 regulates the  expression of autophagy genes".

Line 331, " Figure 4. ERBB2 regulates expression of autophagy genes." To " Figure 4. ERBB2 regulates the expression of autophagy genes.".

Lines 356-357, " on survival probability" to " on the survival probability".

Line 424, "(herceptin)" to "(Herceptin)".

Line 432, " These data suggest that downregulation of ATG12" to " These data suggest that the downregulation of ATG12".

Lines 439-440, " caused by degradation of ERBB2" to " caused by the degradation of ERBB2".

Line 455, " Autophagy plays important roles" to " Autophagy plays important role".

Line 456, " triggers strong interests" to " triggers strong interest".

Line 458, " key roles" to " key role".

Line 479, " autohagy inhibitors" to " autophagy inhibitors".

Line 487, " tissues lead to upregulation of several essential autophagy gene " to " tissues leads to upregulation of several essential autophagy genes ".

Line 495, " to worse outcome to breast" to " to a worse outcome for breast".

Line 497, " Increased level of autophagy" to " Increased levels of autophagy".

Line 526, " and worse outcome to patients" to " and a worse outcome for patients"

4) The “NIC-ERBB2” labeling used (e.g. in Figure 1) to indicate ERBB2 knock-out cells is potentially confusing with cDNA transfer conditions. I recommend spelling out its significance in Figure Legend.

Response: Thank the reviewer, " NIC-Con, Control cells; NIC-ERBB2, ERBB2 knockout cells." has been added in the legends of Figs. 1, 4, 7, S6 and S7

5) It is understood that ATG12 and ATG5 usually form a complex during autophagy, but it is not clear to this reviewer how this complex is recognizable in the presented data? For instance, separate antibody immunoblotting are shown for ATG5 and ATG12 in Figure 4 . Thus, how do the authors know that these proteins are found “in form of an ATG12-ATG5 complex” as indicated in the figure and in the respective legend?  Please explain.

Response:  ATG12 antibody (https://www.cellsignal.com/products/primary-antibodies/atg12-d88h11-rabbit-mab/4180) and ATG5 antibody (2630 (https://www.cellsignal.com/products/primary-antibodies/atg5-antibody/2630)  identify different parts of the ATG12-ATG5 conjugate complex. Furthermore, these two antibodies actually mainly identify the conjugate but not their free forms, respectively. This has been accepted in the autophagy field. A new sentence " Anti-ATG12 (4180) and anti-ATG5 (2630) antibodies mainly identify the ATG12-ATG5 conjugate complex." was added to "2. Materials and Methods" section.

This manuscript is a resubmission of an earlier submission. The following is a list of the peer review reports and author responses from that submission.

Round 1

Reviewer 1 Report

In the present manuscript the authors have investigated the role of ERBB2 in mediating the treatment resistance in breast cancer cells through the induction of an ATG12-dependent autophagy process and the potential efficacy of various therapeutic treatments in these cells, supporting the rationale of ERBB2-targeted therapy. In particular, this manuscript focuses on the effects which the exposure to several anti-ERBB2 treatments may exert in breast cancer.

This investigation appears to be interesting and could have a strong clinical impact as regards the treatment of ERBB2-positive breast tumors with ERBB2-targeted therapies able to overcome drug resistance mechanisms, by improving the effectiveness of anticancer therapies and allowing the selection of patients who may benefit from treatment. This is a useful and original topic that makes the proposed manuscript attractive and worthy of consideration. The article should be of potential interest to a broad readership interested in translational cancer research.

The proposed paper is readily understandable, because it is well-constructed, clear and well described with figures and tables exhaustive and appropriate to the subject matter. The scientific background and aims are clearly explained. The methodology is appropriate to the research field and the conclusions logically follow from the results obtained by performed experiments. The topic falls within the scope of the subject area of the journal. For all these reasons, in my opinion, this manuscript can be considered adequate for the standards required for the publication in Cancers, after minor changes which will help the authors to improve it.

Minor revisions:

1) Rows 232-233 (Results section): Please place correctly this sentence within the legend of Figure 2.

2) Materials and Methods section: please supply city and country for all cited suppliers.

3) Please verify the references #49 and #50 reported in the References section of the manuscript.

4) The authors should quote and discuss the paper by Chen S. et al. (PMID: 26369543), concerning the role of autophagy in the resistance of HER2-positive breast cancer cells to lapatinib.

Author Response

Please see attached letter.

Reviewer 2 Report

In the current research investigation Chen et al. have investigated the role of Atg12 chemo-resistance in context of Erb-b2 receptor in breast cancer. They have used three breast cancer cell lines, MEF, and patient samples to investigate their hypothesis. They used the most acceptable methods in the field to gain their specific objectives. They used chemical inhibition and gene KD and KO, flow cytometry apoptosis assay, and dead cell exclusion assay. They also used ICC to investigate cytosolic and lysosomal LC3-II. They also over-expressed KO genes to exclude potential off target effects in their model. On the other hand the team are also a leader in the field and did current investigation in continious of their previous research. There are some minor points that should be explained or provide data/evidence:

1- It is very important that the respected team show p62 in experiments dealing with autophagy flux.

2- The respected team should have more discussion on the different between triple negative cell line results and other cell line.

3- If the author access to Atg12 protein expression in their human samples, I would very important they provide them in the paper. 

4- I suggest the respected team provide a suggested scheme as a graphical abstract.

Author Response

Reviewer 2

1) It is very important that the respected team show p62 in experiments dealing with autophagy flux.

Response: New data about p62 were added in Figs. 1 (lines 260-267), 3 (lines 306-308) and S1. The methods were also added (line 135-138). Results were presented in lines 236-239 and lines 243-246. We show that HER2 reduces p62 expression indicating autophagy flux.

2) The respected team should have more discussion on the different between triple negative cell line results and other cell line.

Response: This now has been discussed in the Discussion section in the manuscript (lines 497-505).

3) If the author access to Atg12 protein expression in their human samples, I would very important they provide them in the paper. 

Response: Examples of IHC staining of ATG12 proteins in 2 human breast tumors were provided in Fig. S3 and presented in the results section (line 358-359). It shows that ATG12 is expressed in breast tumors.

4) I suggest the respected team provide a suggested scheme as a graphical abstract.

Response: Graphic model was added in Fig. 9 (line 514-520).

Reviewer 3 Report

This manuscript by Chen and coworkers reports data supporting the role of ErbB2 kinase in promoting autophagy in cancer cells, and the potential relevance of ErbB2 inhibitors for blocking this adaptive mechanism and enhance the efficacy of anti-cancer therapies. The presented findings seem to clarify a controversy based on previous data about the actual involvement of ErbB2 in autophagy regulation and also nominate a putative effector of this pathway, ATG12. Although the message of the study can be very important in the field of cancer therapy, certain aspects need to be strengthened and clarified.

Specific comments:

  1. Most of the data shown in this manuscript derive from western blotting experiments analyzing the expression of the autophagy marker LC3B. It is stated in the legend to Fig. 1 that “Data represent the results from at least three independent experiments (the same hereafter)”. In fact, band intensity quantification is shown only for the unique experiments shown in the figures. However, since these results are truly the core of the paper, and the difference in band intensity is barely appreciable in some of the figures, an accurate statistical analysis of these data, based on quantification across several experiments should be provided.
  2. Moreover, simply monitoring autophagy flux by LC3B-II westerns is not sufficient to claim a change in autophagic flux. It is recommended to use at least 2- 3 separate assays for monitoring autophagic flux including confocal imaging to monitor LC3, and western blots for autophagy-specific substrates like p62 (also see Guidelines for the use and interpretation of assays for monitoring autophagy. Autophagy. 2016; 12:1-222).
  3. Intriguingly, there is no evidence of an additive effect of amino acid deprivation with chloroquine (CQ) treatment in Fig. 1B. Could the Authors discuss this?
  4. The impact of ERBB2 knock-out on the formation of fluorescent mRFP-LC3 puncta is not obvious from the analysis of representative fields shown in Fig. 1C. Moreover, the quantification shown in panel D is not very convincing, nor the statistical assessment based on low numerosity and considering the high standard deviation of the values. In order to consolidate these data, I recommend taking in account a double number of cells (N=20 being far too low) and suggest to show more representative images in Suppl. materials.
  5. The interpretation of data shown in Fig. 3B is particularly problematic. ERBB2 is transfected into NIH-3T3 cells in order to test its impact on autophagy, but it is not clear whether this induces any increase in the LC3B marker; if anything, autophagy induction by CQ in these conditions is reduced (40% increase instead of 100% increase in controls, based on the band quantification indicated in the figure). Yet, the treatment with the kinase inhibitor Lapatinib seems to inhibit autophagy both basally and in presence of CQ. It would be important to establish if ERBB2 expression per se is capable of eliciting autophagy. Notably, data shown in panel A may suggest that a relatively low ERBB2 overexpression could be achieved in these cells. It is not clear if in these conditions the kinase is auto-phosphorylated, as observed in overexpressing tumor cells. This could be easily assessed. Moreover, in order to establish the relevance of ErbB2 kinase activity targeted by lapatinib, LC3B band intensity should be eventually compared to that observed in control cells. As stressed above, this information should be acquired based on statistics across more than one experiment.
  6. For experiments shown in Fig. 7A, SKBR3 cells were treated with Taxol alone or its combination with NH4Cl. The descriptive text in Results alludes to a strong impact of ERBB2 knock-out on Taxol-induced autophagy, but actually this is appreciable as described only in the presence of NH4Cl (while the text is not making this explicit). Actually, when performing the assay in absence of NH4Cl (as for all the other experiments in this study), ERBB2 knock-out seems instead to promote taxol-induced autophagy, which is contradictory with the claim of the paper. This issue should be clearly stated in the text and its relevance discussed.
  7. The evidence of ErbB2-ATG12 co-immunoprecipitation shown in Fig. 8C is not clearly described and discussed. What do the authors mean by “in the form of ATG12-ATG5” complex? How exactly was the experiment performed? Where are the complexes supposed to form in the cell? In situ imaging by immunofluorescence may help to support this evidence.
  8. Data shown in Fig. 8D are puzzling. A very strong effect is observed here upon ERBB2 targeting with the antibody trastuzumab. However, this effect is far greater compared to that achieved upon ERBB2 complete knock-out in the same conditions (data shown in Fig. 1B). How can this be explained and what are the implications of this difference?     

Additional issue:

Lines 232-233 have been paginated erroneously, as they probably represent the final part of the legend to Figure 2.

Author Response

Reviewer 3

  •  Most of the data shown in this manuscript derive from western blotting experiments analyzing the expression of the autophagy marker LC3B. It is stated in the legend to Fig. 1 that “Data represent the results from at least three independent experiments (the same hereafter)”. In fact, band intensity quantification is shown only for the unique experiments shown in the figures. However, since these results are truly the core of the paper, and the difference in band intensity is barely appreciable in some of the figures, an accurate statistical analysis of these data, based on quantification across several experiments should be provided.

Response: Bar graphs summary data from 5 independent western blots of LC3B and from 3 independent western blots of p62 were added in Fig. 1 (lines 260-267), and the related supplemental data (uncropped western blots and their intensity quantification.

  • Moreover, simply monitoring autophagy flux by LC3B-II westerns is not sufficient to claim a change in autophagic flux. It is recommended to use at least 2- 3 separate assays for monitoring autophagic flux including confocal imaging to monitor LC3, and western blots for autophagy-specific substrates like p62 (also see Guidelines for the use and interpretation of assays for monitoring autophagy. Autophagy. 2016; 12:1-222).

Response: New data about p62 added in Figs. 1 (lines 260-267), 3 (lines 306-308) and S1. The methods were also added (line 135-138). Results were presented in lines 236-239 and lines 243-246. 

  • Intriguingly, there is no evidence of an additive effect of amino acid deprivation with chloroquine (CQ) treatment in Fig. 1B. Could the Authors discuss this?

Response: It is very likely that in ERBB2 -positive breast cancer cells, amino acid deprivation did not increase autophagy. This is suggested by some of our western blots of LC3B (with and without amino acid starvation) run on a same gel (please see the uncropped western blots related to Fig. 1 in the supplemental data). Furthermore, there was a mistake in Fig. 1D in the previous version of the manuscript, where the bar graphs of LC3 puncta quantification without (left) and with (right) amino acid starvation were accidentally exchanged during the preparation of the manuscript (we thank the reviewer for the comment so that we can notice this mistake). This has been corrected in the current version (Fig. 1B, line 238-241). Since more detailed studies are needed for the role of amino acid starvation in ERBB2-positive cancer cells, we did not emphasize this surprising finding. Therefore, we will not show the data of LC3 puncta with amino acid starvation in this manuscript. Instead, this will be demonstrated in our future publication.

  • The impact of ERBB2 knock-out on the formation of fluorescent mRFP-LC3 puncta is not obvious from the analysis of representative fields shown in Fig. 1C. Moreover, the quantification shown in panel D is not very convincing, nor the statistical assessment based on low numerosity and considering the high standard deviation of the values. In order to consolidate these data, I recommend taking in account a double number of cells (N=20 being far too low) and suggest to show more representative images in Suppl. materials.

Response: As mentioned in the response to Comment # 3, a mistake of Fig. 1D in the previous version was corrected. In the current version (Fig. 1B), more cells (n≥36) were counted (line 239-241) and more images of LC3 puncta were provided in Supplemental Fig. S1A.

  • The interpretation of data shown in Fig. 3B is particularly problematic. ERBB2 is transfected into NIH-3T3 cells in order to test its impact on autophagy, but it is not clear whether this induces any increase in the LC3B marker; if anything, autophagy induction by CQ in these conditions is reduced (40% increase instead of 100% increase in controls, based on the band quantification indicated in the figure). Yet, the treatment with the kinase inhibitor Lapatinib seems to inhibit autophagy both basally and in presence of CQ. It would be important to establish if ERBB2 expression per se is capable of eliciting autophagy. Notably, data shown in panel A may suggest that a relatively low ERBB2 overexpression could be achieved in these cells. It is not clear if in these conditions the kinase is auto-phosphorylated, as observed in overexpressing tumor cells. This could be easily assessed. Moreover, in order to establish the relevance of ErbB2 kinase activity targeted by lapatinib, LC3B band intensity should be eventually compared to that observed in control cells. As stressed above, this information should be acquired based on statistics across more than one experiment.

Response: Thank you to the reviewer. This figure has been updated in the current version (Fig. 3 and the related uncropped western blots in supplemental data) by adding data of EGFR and ERBB2 phosphorylation, effects of EGFR and ERBB2 on autophagy demonstrated by western blots of LC3 and p62 with bar graph of protein band quantification (line 286-294).

  • For experiments shown in Fig. 7A, SKBR3 cells were treated with Taxol alone or its combination with NH4Cl. The descriptive text in Results alludes to a strong impact of ERBB2 knock-out on Taxol-induced autophagy, but actually this is appreciable as described only in the presence of NH4Cl (while the text is not making this explicit). Actually, when performing the assay in absence of NH4Cl (as for all the other experiments in this study), ERBB2 knock-out seems instead to promote taxol-induced autophagy, which is contradictory with the claim of the paper. This issue should be clearly stated in the text and its relevance discussed.

Response: The use of NH4Cl instead of CQ in the experiments with Taxol was originally chosen based on the observation that CQ alone can induce higher cell death than NH4Cl (unpublished previous data and Fig. S5 line 411-413) and higher cell death was observed when combine a chemical drug with CQ than the case with NH4Cl (unpublished previous data). For measuring autophagy by western blot, a lysosomal inhibitor, NH4Cl or CQ, was used to block the degradation of LC3 or p62 in the autolysosomes. An increase in the protein level of LC3 or p62 in the presence of NH4Cl or CQ compared to that in the absence of NH4Cl or CQ indicates a functional autophagy. In the absence of NH4Cl or CQ, an increase of LC3 could be caused by two possibilities including (1) an increase of autophagy (this was the concept/method that was used in the early years of autophagy studies as the methods for autophagy studies were not mature, and "an increase in LC3 protein indicates an increase of autophagy" was widely accepted during that period); (2) an inhibition of autophagy. When autophagy degradation was inhibited in the autolysosome by some treatments (here, ERBB2 downregulation), the protein level of LC3 may be high in the absence of NH4Cl or CQ and the protein level of LC3 is low in the presence of NH4Cl or CQ. Thanks to the landmark paper published by Noboru Mizushima and Tamotsu Yoshimori (Autophagy 3:6, 542-545, 2007), now it is widely accepted that "an increase in LC3 protein" in the absence of a lysosomal inhibitor may not indicate an increase in autophagy or the existence of a functional autophagy.

  • The evidence of ErbB2-ATG12 co-immunoprecipitation shown in Fig. 8C is not clearly described and discussed. What do the authors mean by “in the form of ATG12-ATG5” complex? How exactly was the experiment performed? Where are the complexes supposed to form in the cell? In situ imaging by immunofluorescence may help to support this evidence.

Response: Since it is known that ERBB2 antibody can lead to ERBB2 protein degradation. The demonstration of IP data showing that ERBB2 interaction with ATG12 was used to propose a possible mechanism for ATG12 degradation by ERBB2 antibody treatment. It is well known that ATG12 and ATG5 usually form a conjugate in the form of ATG12-ATG5 during autophagy. Presentation of the ATG12-ATG5 complex in western blot of ATG12 or ATG5 has widely been used in most studies in the autophagy field. This has been mentioned in the legend of Figure 4 in the current manuscript as the reviewer suggested (line 337-340).

  • Data shown in Fig. 8D are puzzling. A very strong effect is observed here upon ERBB2 targeting with the antibody trastuzumab. However, this effect is far greater compared to that achieved upon ERBB2 complete knock-out in the same conditions (data shown in Fig. 1B). How can this be explained and what are the implications of this difference?

Response: Although ERBB2 antibody treatment can led to ERBB2 degradation, this treatment may not exactly mimic ERBB2 knockout. ERBB2 antibody treatment can exert other effects. For example, herceptin/trastuzumab can increase the phosphorylation of EGFR (Gijsen M et al. Plos Biology 2016; 14(3):e1002414--Reference 38 in the manuscript) leading to autophagy inhibition as demonstrated by our previous study (Chen Y et al. Autophagy 2016; 12(6):1029-1046---Reference 12). This has now been discussed in the last sub-section of the Results (3.5.) as the reviewer suggested (line 431-435).

Additional issue:

Lines 232-233 have been paginated erroneously, as they probably represent the final part of the legend to Figure 2.

Response: This has been corrected in the current manuscript (line 302-308).